# W Chromosome Evolution by Repeated Recycling in the Frog *Glandirana rugosa*

Mitsuaki Ogata [1,*], Foyez Shams [2], Yuri Yoshimura [3], Tariq Ezaz [2] and Ikuo Miura [2,4,*]

1   Preservation and Research Center, 155-1 Asahi Ward, Yokohama 241-0804, Japan
2   Institute for Applied Ecology, University of Canberra, Canberra, ACT 2617, Australia;
    foyez.shams@canberra.edu.au (F.S.); tariq.ezaz@canberra.edu.au (T.E.)
3   Department of Biology, Faculty of Science, Kyushu University, Fukuoka 819-0395, Japan;
    yuriyoshimura530@gmail.com
4   Amphibian Research Center, Hiroshima University, Higashi-Hiroshima 739-8526, Japan
*   Correspondence: zvp06246@nifty.ne.jp (M.O.); imiura@hiroshima-u.ac.jp (I.M.); Tel.: +81-82-424-7323 (I.M.)

**Abstract:** The Y or W sex chromosome of a heteromorphic pair is usually heterochromatinised and degenerated. However, whether chromosome degeneration constantly proceeds toward an extreme end is not fully understood. Here, we present a case of intermittent evolution of W chromosomes caused by interpopulation hybridisation in the Japanese soil-frog, *Glandirana rugosa*. This species includes two heteromorphic sex chromosome systems, which are separated into geographic populations, namely the XY and ZW groups. In this study, to uncover the evolutionary mechanisms of the heterogeneous W chromosomes, we genetically investigated the geographic differentiation of the ZW populations along with the closely located XY populations. Analysis of mitochondrial *cytochrome b* sequences detected three distinct clades, named ZW1, ZW2, and ZW3. High throughput analyses of nuclear genomic DNA showed that autosomal alleles of XY populations were deeply introgressed into the ZW3 sub-group. Based on the genotypes of sex-linked single nucleotide polymorphisms, W-borne *androgen receptor* gene expression, and WW developmental mortality, we concluded that the X chromosomes were recycled to W chromosomes. Upon inclusion of two cases from another group, Neo-ZW, we observed that the X chromosomes were recycled independently at least four times to the new W chromosomes: a repetition of degeneration and resurrection.

**Keywords:** ZW; XY; chromosome degeneration; chromosome resurrection; geographic differentiation

## 1. Introduction

Sex chromosomes are originally homomorphic in both sexes and can be shifted to a heteromorphic state in either sex, which involves the heterochromatinisation and degeneration of the Y or W chromosome [1–3]. In mammals, the degeneration process is accelerated by an inversion on the Y chromosome, which inhibits recombination with the corresponding region of the X chromosome, thus forming a stratum on the sex chromosomes [4,5]. In addition, an old Y chromosome rarely reaches an extreme end and becomes extinct, as observed in three rodents [6–9]. In contrast, although avian ZW sex chromosomes share their evolutionary origin (which occurred 100 million years ago) for the same homomorphic pair, the W chromosome degeneration depth differs in the two major taxa, Neognathae and Palaeognathae. In the Neognathae lineage that includes chickens, the W chromosomes are tiny, highly differentiated, degenerated, and heterochromatinised [10,11], whereas in the Palaeognathae lineage that includes Emu, these continue to conserve similar sizes keeping lots of genes to the Z chromosomes [10,12–14]. The degeneration process of Y or W chromosomes does not always proceed at a constant rate, being extremely rapid in some cases (or initially after recombination arrest), while quite slow in others (or later) [15]. What controls the degeneration rate? Does it depend on chromosomal rearrangements such as inversion, or the time elapsed since recombination arrest between the sex chromosome pair?

In 2012, we encountered an unexplainable case of W chromosome heterogeneity in the Japanese soil-frog, *Glandirana rugosa*, suggesting independent evolution of the W chromosomes in two different populations (Niigata and Kanazawa), whose geographic locations are approximately 260 km apart, despite belonging to the same geographic group [16]. The frog species comprises six geographic groups based on sex determination, sex chromosomes, and mitochondrial haplotypes (Figure 1) [17–19]. Of these, three groups possess heteromorphic sex chromosomes of the ZZ-ZW or XX-XY type and are termed as ZW group in north-eastern Japan, Neo-ZW group in west-central Japan, and XY group in east-central Japan [17–19]. The sex chromosomes in all three groups are represented by homologous chromosome 7 in the 13 haploid complements: the X and W chromosomes are metacentric, sharing one origin, while the Y and Z chromosomes are subtelocentric, sharing the other origin [16–18]. The Neo-ZW group has a recent origin from the XY group and is divided into two sub-groups of Neo-ZW1 and Neo-ZW2 based on the mitochondrial haplotypes: the latter is proved to be derived from more recent hybridisation between the Neo-ZW1 sub-group and the XY group [20]. It has been reported in previous studies that the artificially constructed WW embryos of the ZW group die at an early developmental stage due to the presence of lethal genes degenerated on the W chromosomes, but the stages of developmental arrest as well as the symptoms of morphological anomalies differ in the two populations, Niigata (N) of Niigata Prefecture and Kanazawa (K) of Ishikawa Prefecture, occurring at 10 days post fertilisation (dpf) of edemata and 40 dpf of underdevelopment, respectively. In addition, it was reported that artificially constructed hybrid $W^N W^K$ embryos of the two populations were completely viable. This depicted that the W chromosomes do not share the recessive lethal genes, thus proving that they evolved independently, but not serially from the original W chromosome.

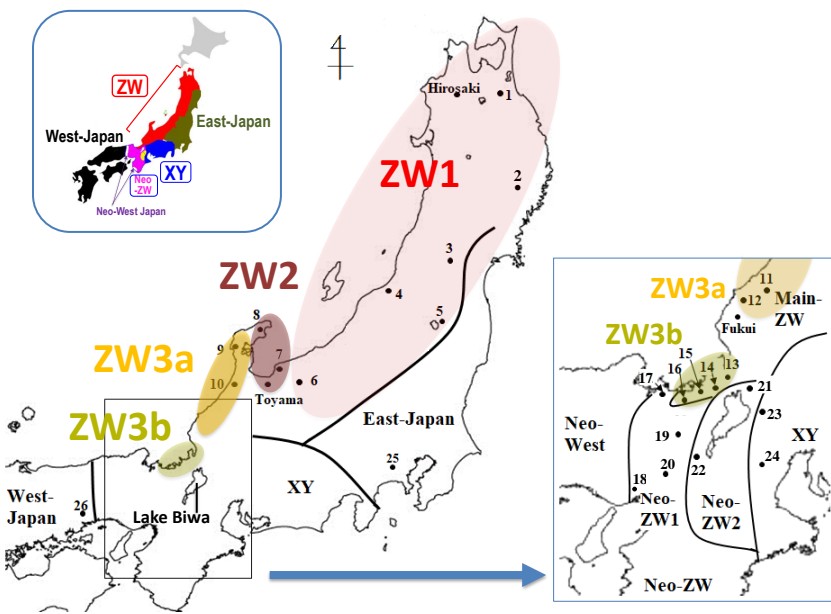

**Figure 1.** Map showing locations of 26 populations in *Glandirana rugosa* used in this study. The populations of ZW group are divided into three sub-groups closed in colored circles. Dots with numbers indicate 26 populations used in this study and three cities (with names of Hirosaki, Toyama and Fukui, shown in the study of Miura et al. [16]). The major six geographic groups are shown at top left and magnification of the central Japan area is shown at bottom right.

In this study, to uncover the evolutionary mechanisms of W chromosome evolution in the ZW group of *G. rugosa*, we conducted analyses of its mitochondrial as well as nuclear genomic DNA.

## 2. Materials and Methods

### 2.1. Experimental Animals

A total of 195 *G. rugosa* frogs were collected from 26 populations in the northern and central regions of the Japanese archipelago (Table 1; Figure 1). The sex of the specimens was determined by inspection of the gonads after euthanasia. Animal care and experimental procedures were approved by the Committee for Ethics in Animal Experimentation at Hiroshima University (Permit Number: G18-2-3).

**Table 1.** Sex-linked genotypes and mitochondrial haplogroups in 24 populations of ZW, Neo-ZW and XY groups and 2 populations of East-Japan and West-Japan groups.

| Population Number | Population (City) | Prefecture | No. of Frogs Examined Male | No. of Frogs Examined Female | Heterogametic Sex Based on SF1 Genotype | Microsatellite Allele of *Sox3*-UTR on W or X Chromosome | Expression of *Androgen receptor* Gene on W or X Chromosome | *Cytochrome b* Haplotype (Accession No.) | *Cytochrome b* Haplogroup |
|---|---|---|---|---|---|---|---|---|---|
| 1 | Towada | Aomori | 6 | 8 | Female | 230 | NE | L671792 | **ZW1** |
| 2 | Toono | Iwate | 0 | 1 | Female | NE | − | L671793 | **ZW1** |
| 3 $ | Yamagata | Yamagata | 7 | 3 | Female | 230 | NE | L671794 | **ZW1** |
| 4 * | Niigata | Niigata | 0 | 1 | Female | NE | − * | L671795 | **ZW1** |
| 5 | Inawashiro | Fukushima | 4 | 7 | Female | 230 | NE | L671793 | **ZW1** |
| 6 | Oomachi | Nagano | 2 | 1 | Female | 230 | NE | L671792 | **ZW1** |
| 7 $ | Kurobe | Toyama | 9 | 5 | Female | 223 | − | L671796 | ZW2 |
| 8 | Suzu | Ishikawa | 3 | 4 | Female | 275 | − | L671797 | ZW2 |
| 9 $ | Tomiki | Ishikawa | 1 | 3 | Female | 266/284 | NE | L671798 | ZW3a |
| 10 * | Kanazawa | Ishikawa | 0 | 1 | Female | NE | − * | L671799 | ZW3a |
| 11 $ | Yamanaka | Ishikawa | 1 | 2 | Female | 230 | − | L671800 | ZW3a |
| 12 $ | Awara | Fukui | 3 | 6 | Female | 248 | +3/−3 | L6717801 | ZW3a |
| 13 | Tsuruga | Fukui | 3 | 7 | Female | 240 | +6/−1 | L6717804 | ZW3b |
| 14 $ | Wakasa | Fukui | 2 | 8 | Female | 240 | + | L6717802 | ZW3b |
| 15 | Obama | Fukui | 3 | 7 | Female | 240 | +6/−1 | L6717803 | ZW3b |
| 16 | Kamikato | Fukui | 4 | 6 | Female | 240 | + | L6717804 | ZW3b |
| 17 | Maizuru | Kyoto | 1 | 4 | Female | 240 | +3/−1 | L6717808 | Neo-ZW1 |
| 18 * | Sanda | Hyogo | 0 | 1 | Female | 240 | NE | L6717809 | Neo-ZW1 |
| 19 *$ | Kashiwabara | Kyoto | 8 | 4 | Female | 240 | NE | | NE (Neo-ZW1) |
| 20 *$ | Osaka | Osaka | 5 | 6 | Female | 240 | + | | NE (Neo-ZW1) |
| 21 *$ | Kinomoto | Shiga | 4 | 8 | Female | 222/232 | + | L6717805 | Neo-ZW2 |
| 22 *$ | Kyoto | Kyoto | 3 | 3 | Female | 248 | + | L6717807 | Neo-ZW2 |
| 23 *$ | Sekigahara | Gifu | 3 | 3 | Male | 222/232 | + | L6717805 | **XY** |
| 24 *$ | Kameyama | Mie | 5 | 10 | Male | 222/232/240/248 /250 | + | | **XY** |
| 25 | Isehara | Kanagawa | 0 | 1 | Homo # | NE | NE | L6717810 | East-J |
| 26 | Kamigoori | Hyogo | 0 | 1 | Homo # | NE | NE | L6717806 | West-J |
| | Total | | 79 | 116 | | | | | |

NE, not examined, #, sex linked genes are homozygous in both sexes, *, heterogametic sex, microsatellite allele of *Sox3*-upstream region and expression of *Ar* gene on W chromosome or X chromosome are previously reported [20], $, used for DArT analysis, −, *Ar* is not expressed; +, expressed.

### 2.2. Identification of Heterogametic Sex

Total genomic DNA was extracted from the blood cells using a DNeasy blood and tissue kit (QIAGEN, Hilden, Germany) according to the manufacturer's instructions. The heterogametic sex of the frogs was determined based on the genotypes of two sex-linked genes, *ADP/ATP translocase* and *steroidogenic factor 1*, according to a previously described genotyping protocol [20]. We previously reported the heterogametic sex of *G. rugosa* frogs collected from nine populations in the Kinki district of Central Japan, indicated by * in Table 1 [20,21].

### 2.3. Amplification of Mitochondrial DNA and Construction of Phylogenetic Trees

A 918-base pair (bp) partial fragment of the mitochondrial *cytochrome b* gene was amplified by polymerase chain reaction using the following primer set: forward 5′-TYA CCG GCC TAT TCC TAG C-3′ and reverse 5′-CCTARKGTGGGDAYAAGAAGGAC-3′. The *cytochrome b* sequence of one female from each of the 16 populations was determined (Table 1) using an ABI PRISM 370 genetic analyser (Thermo Fisher Scientific, Waltham, MA, USA) according to the manufacturer's instructions. DNA sequences were aligned

using MEGA 6 [22] and deposited in the DDBJ Data Libraries (accession number LC671792-671810). For construction of gene trees, the best substitution model for each gene was selected using Kakusan4 [23] based on the Akaike information criterion (AIC). Phylogenetic trees were constructed on the basis of the maximum-likelihood method (ML) and Bayesian inference (BI) using TREEFINDER ver. March 2011 (Munich, Germany) [24] and MrBayes ver. 3.2.1 [25]. For the ML tree, nonparametric bootstrap analysis was conducted using 1000 replicates; branches with a bootstrap value of 70% or greater were regarded as significantly supported. In the BI analysis, two independent runs of four Markov chains were conducted for 10 million generations (sampling frequency of one tree per 100 generations) of which the first three million generations were discarded as burn-in. The convergence of the parameters was checked using Tracer ver. 1.5 [26]. A Bayesian posterior probability of 0.95 or greater was accessed as significant support. To construct the phylogenetic trees, the sequence data of *Pelophylax nigromaculatus*, *Pelophylax fukienensis*, and *Pelophylax lessonae* were used as outgroups (AB980710/AB980717, AB980780, AB980716).

## 2.4. RT-PCR of Sex-Linked Androgen Receptor

To investigate the expression of Androgen Receptor (Ar) located on the W chromosome of frogs collected from P12-17 populations in the Fukui as well as Kyoto Prefectures, total RNA was extracted from the fingers of frogs using SV Total RNA Extraction Kit (Promega, Madison, WI, USA) according to the manufacturer's instructions. cDNA was synthesised using a PrimeScript 1st strand cDNA synthesis kit (TaKaRa, Kyoto, Japan) with a $(dT)_{20}$ oligonucleotide primer. RT-PCR was performed according to a previously described method [21].

## 2.5. Microsatellite DNA Amplification of Sex-Linked Sox3 Upstream Region

Microsatellite DNA from the sex-linked *Sox3* upstream region was amplified from female and male frogs according to a previously described method [20]. Two kinds of fluorochrome (5Hex and 6 FAM) were attached to the 5′ end of the forward primers. Electrophoresis of the PCR product was performed using an ABI Prism 310 genetic analyser with GS500 size marker. GeneScan2 (Thermo Fisher Scientific, Waltham, MA, USA) was used for genotyping.

## 2.6. Genome Analysis to Determine Population Genetic Structure

To investigate the genetic structures of populations, genotyping-by-sequencing (GBS) was performed using diversity arrays technology sequencing (DArTseq) as described in [20]. DArTseq uses genome complexity reduction coupled with high-throughput next generation sequencing (NGS) to identify alleles displaying single nucleotide polymorphisms (SNPs) [27]. A total of 76 frogs from 12 populations (P3, 7, 9, 11, 12, 14, 19, 20–24) were genotyped; those with asterisks in Table 1 were obtained from a previous study [20]. From a total of 89,468 SNP sites identified by DArT sequences, 1626 autosomal SNPs were identified with no missing alleles across all samples (Tables S1 and S2). To characterise the genetic relationships amongst the 76 frogs, STUCTURE (ver. 2.3) [28] was run on the 1626 sites for 10,000 Markov chain Monte Carlo cycles following 10,000 burns in cycles, using an admixture model with independent allele frequencies. Ten replications were performed for each K, in the range K = 1–10, and the optimal K was estimated using STRUCTURE HARVESTER [29].

Phylogenetic analysis was conducted using the 1626 sites. Euclidean distances among populations were calculated using dartR [30]. The phylogenetic tree was constructed on the basis of the neighbour-joining method using PHYLIP [31]. Bootstrap analysis was conducted using 1000 replicates.

## 2.7. Genome Analysis to Determine the Genome Structure of Sex Chromosomes

A total of 220 sex-linked SNPs were used to investigate the genome structure of the sex chromosomes (Tables S3 and S4). The sex-linked alleles were identified based on data

from our previous study [20]. Call rates of the 220 SNPs in the 76 frogs were above 0.8. To characterise the genome structure of the sex chromosomes, STUCTURE was run on 220 sites under the same conditions as that used for autosomal alleles. ZZ homogametic frogs were used to characterise Z chromosomes, while ZW and XY heterogametic frogs were used to characterise the ZW and XY chromosomes, respectively. Phylogenetic analysis based on 220 sex-linked SNPs was performed as on the autosomal SNPs.

## 3. Results

### 3.1. Identification of Three ZW Sub-Groups Based on Mitochondrial DNA Sequence

A total of 13 haplotypes of *cytochrome b* were identified in 16 frogs from 16 populations (P1–16) belonging to the ZW group (Table 1). Based on ML or Bayesian methods (Bayes), the haplotypes were found to constitute three distinct clades (ZW1, ZW2, and ZW3). Of these, ZW1 and ZW2 were significantly supported by both methods, whereas ZW3 was not supported by the Bayesian method (BPB = 0.79) (Figure 2). The ZW3 haplotypes were further divided into two subclades, ZW3a and ZW3b.

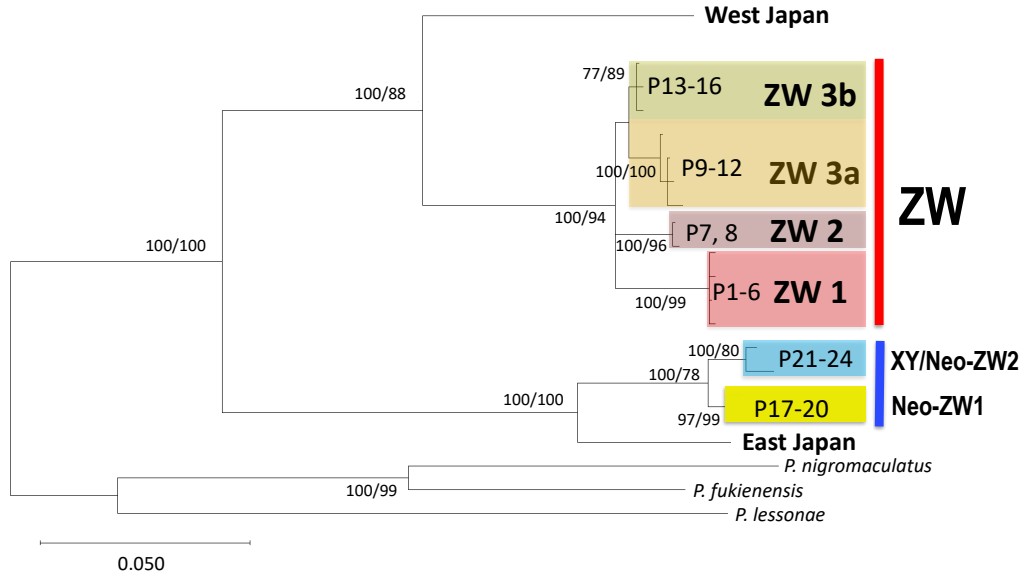

**Figure 2.** Bayesian tree based on 918 bp partial sequences of mitochondrial *cytochrome b*. Numbers at each node are bootstrap probability calculated by maximum likelihood method and Bayesian posterior probability by Bayesian method.

### 3.2. Three ZW Sub-Groups Based on Nuclear Genomes

To confirm the three ZW sub-groups identified on the basis of mitochondrial haplotypes, we investigated the nuclear genomes using diversity arrays technology sequencing (DArTseq). The neighbour-joining trees based on 1626 autosomal SNPs and 200 sex-linked (XY or ZW chromosomes and ZZ chromosomes) SNPs identified in 76 frogs from 12 populations (superscript "$" on the right of the numbers in Table 1; Tables S1 and S2) clearly separated the sub-groups of ZW1, ZW2 and ZW3a from the XY and Neo-ZW groups, but not the ZW3b, which was perfectly located close to the latter (Figure 3). ZW3a did not constitute any monophyly, sharing branches with the ZW2 sub-group.

### 3.3. Population Genetic Structure

To investigate the genetic structure of the ZW populations, we conducted STRUCTURE analysis based on the same set of autosomal SNPs as those used in the phylogenetic tree (Figure 4; Figures S1 and S2). The delta K value was the largest (358.15) in k = 2, 25.26 in k = 3, and 5.82 in k = 4. The K-3 map clearly identified three distinct clusters, XY group (blue), Neo-ZW1 sub-group (yellow), and ZW1 sub-group (red) (Figure 4): XY and ZW1 evolved first [17] and Neo-ZW1 was secondarily derived from the XY group [21]. ZW3a

was found to be genetically composed of three different clusters: blue, yellow, and red, and ZW3b was almost dominated by Neo-ZW1 genome clusters (yellow) (Figure 4, K = 3).

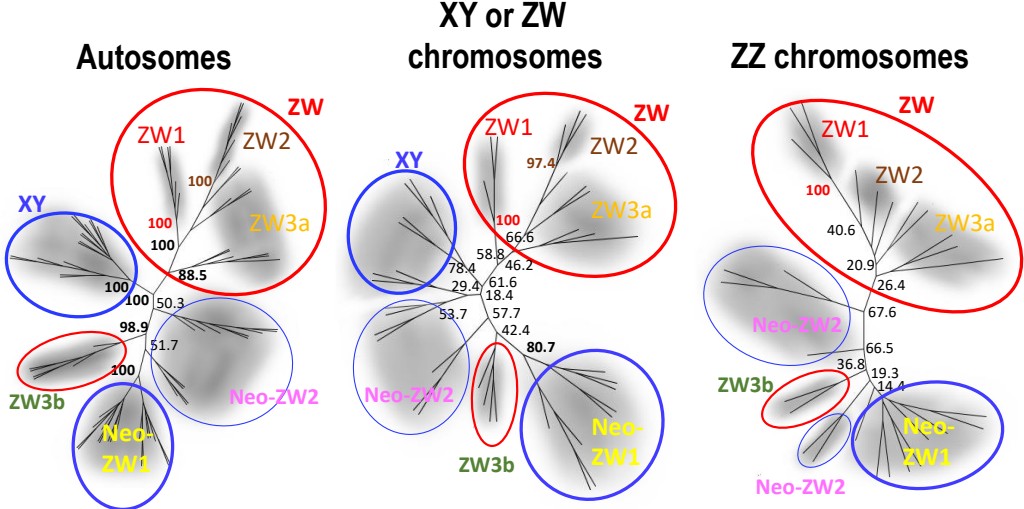

**Figure 3.** Neighbour-joining trees based on SNPs of autosomes, XY or ZW chromosomes, and ZZ chromosomes. Bootstrap probabilities are put on the major nodes. The coloured outlines of circles indicate two major mitochondrial haplogroups of ZW group (red) and XY group (blue). Abbreviations are the same as those of Figure 2.

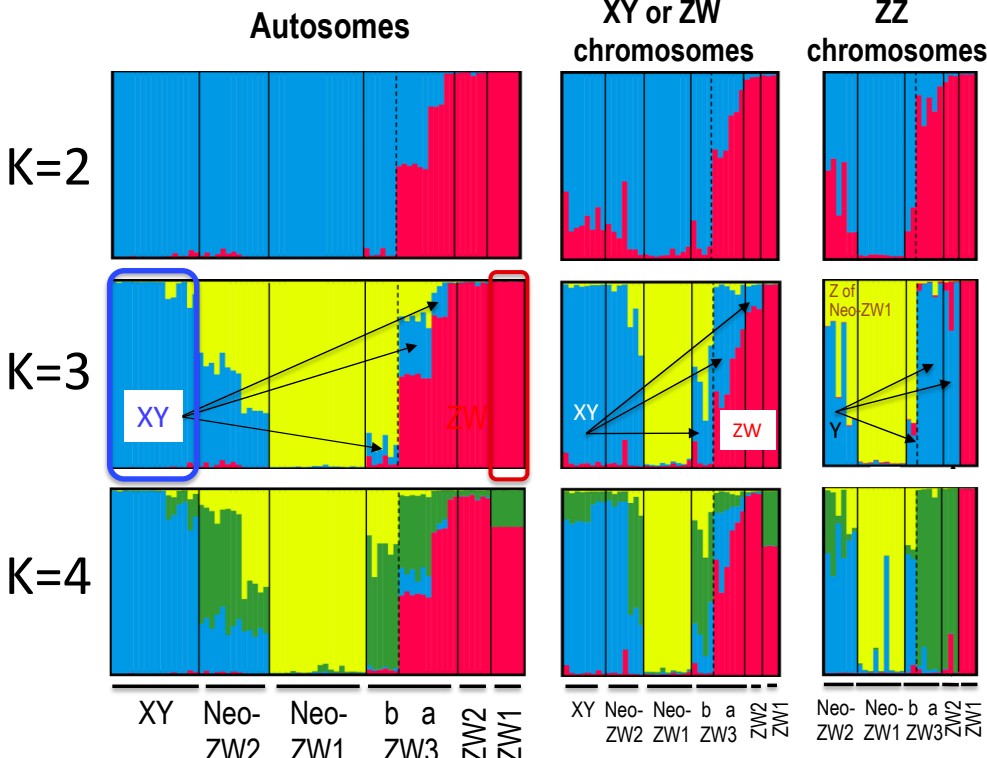

**Figure 4.** Histograms of STRUCTURE assignment test for autosomal 1626 SNPs and 220 sex-linked SNPs. K values (No. of clusters) are 2, 3 and 4. The populations used for the analysis and their locations are shown in Table 1 and Figure 1. The sex-linked SNPs of heterogametic XY males and ZW females were used for investigating genomic structures of ZW and XY chromosomes, while homogametic ZZ males were used for those of Z chromosomes. Arrows indicate introgressions of genomes from XY group to ZW group. Clusters of XY group and ZW1 sub-group are boxed in K = 3 map (autosomes).

### 3.4. Genomic Structure of Sex Chromosomes

To investigate the genomic structure of the sex chromosomes in the ZW populations, separate allele distributions were created for homogametic ZZ males as well as for heterogametic ZW females and XY males in structure maps of 220 sex-linked SNPs (Tables S3 and S4; Figures 4, S1 and S2). Delta K values for heterogametic frogs were 24.60 (k = 2), 17.21 (k = 3), and 1.38 (k = 4), while those for homogametic ZZ frogs were 35.21 (k = 2), 21.94 (k = 3), and 6.95 (k = 4). In the K3 map of heterogametic frogs, XY alleles (blue) constituted half or less of ZW females of the ZW3a or ZW3b sub-groups. In the homogametic ZZ frog map, Y alleles (blue indicates Y alleles while yellow indicates Z alleles from Neo-ZW1 in ZZ males of Neo-ZW2, because this sub-group has a more recent origin at hybridisation between XY and Neo-ZW1, see [20]) constituted one-third of ZZ males from ZW3b, and almost all of ZZ males from ZW3a and ZW2 (arrows in K = 3 map of ZZ chromosomes in Figure 4).

### 3.5. Expression of the Ar Gene on W Chromosome

The *Ar* gene is sex-linked in ZW and XY groups [32,33]: Z-, Y-, or X-borne *Ar* are normally expressed, whereas W-borne *Ar* is not or very faintly expressed [16,34]. To characterise the W chromosomes, we examined *Ar* expression in ZW females from the ZW as well as Neo-ZW groups. *W-Ar* was not expressed in any of the 15 ZW females from the ZW1 or ZW2 sub-groups (P2, P4, P7, and P8) (Table 1). In contrast, it was normally expressed in 20 out of the 21 ZW females from Neo-ZW populations (P17, P20–P22) and 29 out of 37 ZW females of the ZW3 sub-group (P10–P16) (Table 1).

### 3.6. Microsatellite Locus of Sex-Linked Sox3 Upstream Region

Haplotypes of the *Sox3* upstream region, another sex-linked locus, were investigated. A single allele (230), which was characterised by all ZW females of the ZW1 sub-group (P1–P6) (Table 1), was found in one population (P11) of the ZW3a sub-group. In contrast, a single allele (240), which was characterised by all ZW females of the Neo-ZW1 sub-group (P17–20), was found in four populations (P13–16) of the ZW3b sub-groups and in XX females of the XY group (P24). Allele 248, which was identified in one population (P12) of the ZW3a sub-group, was in common with the females of the Neo-ZW2 sub-group (P22) and XY group (P24). These alleles support X chromosome introduction from XY into ZW3 populations. Another four alleles were unique in the remaining population (P9) of the ZW3a sub-group and two populations (P7 and P8) of the ZW2 sub-group (Table 1).

## 4. Discussion

### 4.1. Geographic Differentiation of the Three ZW Sub-Groups

The Japanese frog *G. rugosa* has two heteromorphic sex chromosome systems of XX-XY and ZZ-ZW. Both sex chromosomes are represented by homologous chromosome 7 in the complements. In our previous study [16], we encountered an unexplainable case of W chromosome differentiation: the W chromosomes from two different populations did not share the degenerated lethal genes to cause WW embryo mortality. To uncover the mechanisms of W chromosome heterogeneity, we conducted this study and found out the following.

First, we elucidated that the ZW group bearing ZZ-ZW heteromorphic sex chromosomes comprises three sub-groups (ZW1, ZW2, and ZW3, in the order from north to south geographic region), based on haplotypes of the mitochondrial *cytochrome b* gene (Figure 2). The ZW3 sub-group can be further separated into smaller sub-groups of ZW3a (P9–P12) and ZW3b (P13–P16). Geographically, the boundary between the ZW1 and ZW2 sub-groups, which is located between P6 (Ohmachi) and P7 (Kurobe), almost coincides with the historical boundary between the east and west region of the Japanese mainland, called the Ohi-Itoigawa line [35]. Thus, it is plausible that the ZW1 and the other ZW2–ZW3 sub-groups were separated from each other (and thus genetically differentiated) by a large geographic event in the past that isolated the Eastern and Western Japanese mainland (Figure 5). In addition, according to the geological history of the Hokuriku region, the

Ishikawa, Fukui, and Toyama areas (corresponding to the ZW2 and ZW3a areas in Figure 1) were repeatedly submerged in the sea during the late Pliocene and middle Pleistocene periods [35]. Such submergence may have induced further geographic isolation of the ZW populations and resulted in the formation of the three sub-groups, ZW1, ZW2, and ZW3. Subsequent to the ZW3 sub-group and XY group coming into contact and hybridising in the southern Fukui region (corresponding to the ZW3b region in Figure 1), the ZW3 group might have been divided into ZW3a and ZW3b by sea transgression (Figure 5A–D).

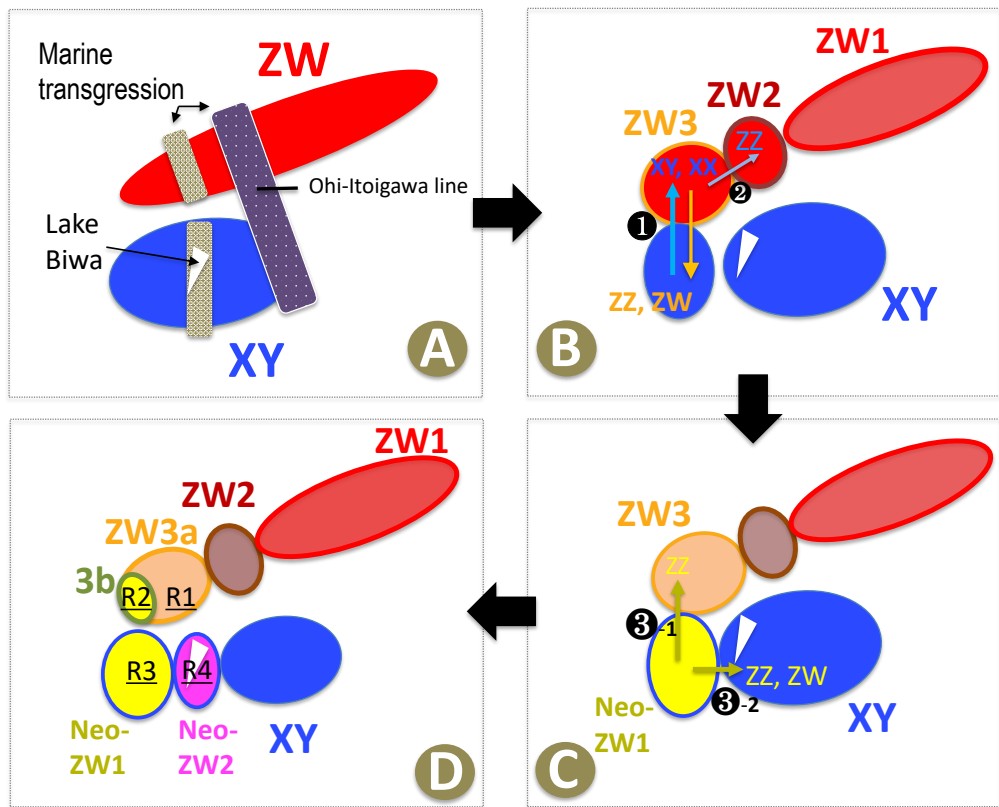

**Figure 5.** Assumed evolutionary process of geographic differentiation of the ZW and Neo-ZW groups and evolution of the W chromosomes in *Glandirana rugosa* (**A**–**D**). Arrows indicate four major emigrations of populations between XY and ZW groups and one emigration from ZW3a to ZW2. R1–R4 denote four times recycling of the W chromosomes from X chromosomes.

*4.2. W Chromosome Heterogeneity as a Result of Repeated X Chromosome Recycling*

Second, we found that the sex chromosomes of the ZW (three sub-groups) and Neo-ZW groups (two sub-groups) were genomically classified into six types of Z chromosomes ($Z^1$, $Z^2$, $Z^{3a}$, $Z^{3b}$, $Z^{N1}$, and $Z^{N2}$) and five types of W chromosomes ($W^1$, $W^{3a}$, $W^{3b}$, $W^{N1}$, and $W^{N2}$), respectively, based on sex-linked SNPs and *Sox3* upstream microsatellite, *W-Ar* expression, and WW embryo mortality (Figure 6). The evolutionary mechanisms of the Z and W chromosomes in the Neo-ZW1 and Neo-ZW2 sub-groups have been described in our previous studies [20,21]. In this study, the $W^1$ chromosomes in the ZW group were found to be shared by the ZW1 and ZW2 sub-groups: *W*-borne *Ar* was not expressed, and WW embryos died at 10 dpf with oedema (L10 in Figure 6). On the other hand, their Z chromosomes are not shared: the $Z^2$ chromosomes of ZW2 are almost entirely replaced with Y chromosomal segments of the XY group, as ZZ clusters of ZW2 sub-group are mostly indicated in blue, which are the Y chromosomal segments of Neo-ZW2 in K3 map of Figures 4 and 6). Although autosomal SNPs did not indicate genomic introgression of the XY group into the ZW2, the genomic region of the Y chromosomes might have been introduced via XY males following separation from ZW1, and then may have been replaced with the $Z^1$ chromosome (Figures 4, 5B-❷ and 6). The genetic structures of the ZW sex

chromosomes differ from each other between the ZW3a and 3b sub-groups: the major part of W$^{3a}$ chromosome mostly comprises the original W$^1$ chromosome except for the *Ar* or lethal genes, whereas W$^{3b}$ mostly comprises parts of X chromosomes, as ZZ clusters of ZW3b are mostly indicated in yellow (Z$^{Neo1}$) while ZW clusters are equally both in yellow and blue showing that the W$^{3b}$ chromosomes originated from X chromosomes (blue) (in K3 map of Figures 4 and 6). W$^{3ab}$-borne *Ar* was or was not expressed, and W$^{3a}$W$^{3a}$ embryos died at 40 dpf owing to underdevelopment (L40). Experimental data for the development of W$^{3b}$W$^{3b}$ embryos is not available. Z$^{3a}$ chromosomes of ZW$^{3a}$ are almost entirely replaced with Y chromosomes (shown in blue in K3 map of Figures 4, 5B-❶ and 6), while major parts of Z$^{3b}$ have accepted introgression from Z$^{N1}$ of Neo-ZW1 sub-groups (shown in yellow in K3 map of Figures 4, 5C-❸-1 and 6).

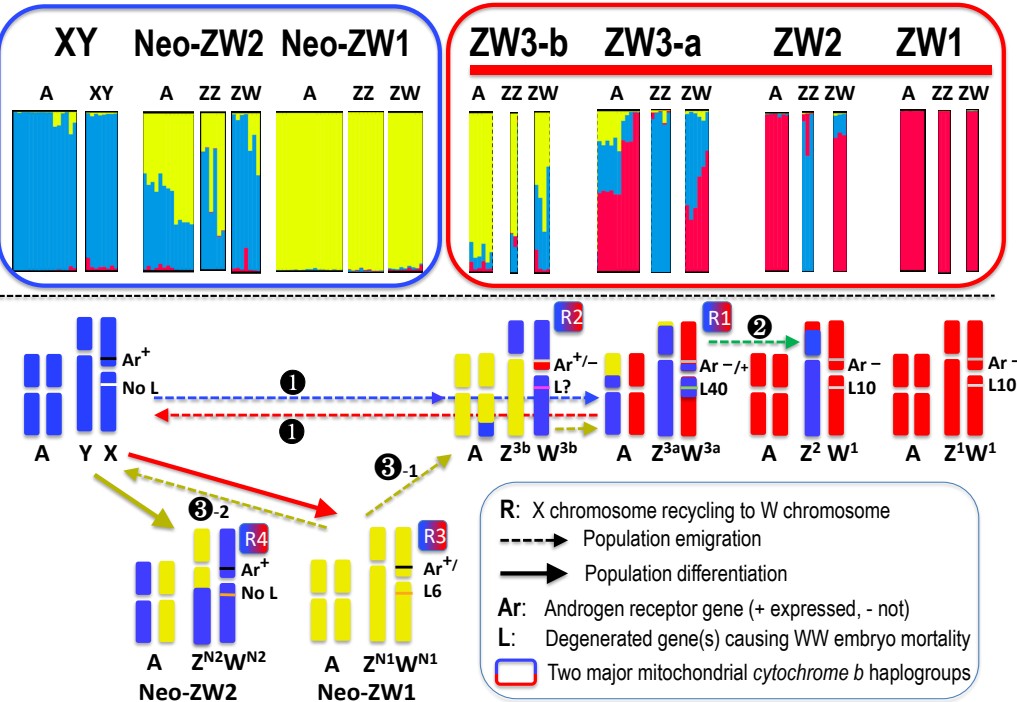

**Figure 6.** Highlights of autosomal and sex-linked clusters selected from STRUCTURE histograms of the geographic groups and sub-groups (upper) and the assumed genomic structures of autosomes and sex chromosomes (lower). Abbreviations are shown in the box at bottom. Six kinds of Z chromosomes and five kinds of W chromosomes were identified. Population emigrations are indicated by dotted lines with 1, 2, 3$^{-1}$ and 3$^{-2}$. Four times reconstruction of the W chromosomes are denoted with R1–R4.

How were the W$^{3ab}$ chromosomes of the ZW3 sub-group reconstructed from the X chromosomes without replacing the mitochondrial haplogroups? Normally, mtDNA as well as W chromosomes are maternally inherited. Therefore, it has been predicted that patterns of phylogenetic differentiation in mtDNA as well as W chromosomes occur in parallel, as reported in avian species [36,37]. In this study, we assumed that the X chromosomes may have been introduced via XY males of the XY group that emigrated into the ZW3 sub-group, without any replacement of mitochondrial haplotypes. A cross between ZW females and XY males would have produced WX hybrid females, in which the W and X chromosomes had undergone a normal recombination because of their identical morphology (metacentric) [38]. The W-X recombination was subsequently confirmed in artificially designed WX hybrid females where chiasmata were observed along the WX bivalent chromosomes [39]. The W-X recombination may have replaced the original W chromosomes, bearing degenerated lethal genes and *Ar*, with those bearing the active form of the genes, finally leading to the heterogeneity of W chromosomes in the ZW group. Upon inclusion of the cases of the Neo-ZW group in our previous study [21], we have concluded

that in the past, X chromosomes derived from the XY group had independently recycled into the new W chromosomes at least four times: W chromosomes of Neo-ZW group (Neo-ZW1 and Neo-ZW2) on the one hand (R3 and R4 in Figures 5 and 6), and W chromosomes of ZW group ($ZW^{3a}$ and $ZW^{3b}$) on the other hand (R1 and R2 in Figures 5 and 6). Recycling revived the W chromosomes by removing degenerated genes such as the lethal recessive genes that arrest development at an early stage as well as the nonactive *Ar*. Thus, X chromosome recycling to the W chromosome can explain the nature of W chromosome heterogeneity in the ZW group that was mentioned in our previous study [16]. These results reveal that the W chromosomes in *G. rugosa* repeated the cycle of degeneration and resurrection, causing intermittent degeneration, by introducing X chromosomes from XY populations of other geographic regions.

### 4.3. Another Model of the W Chromosome Evolution

In this study, our theory of W chromosome reconstruction in the ZW group is based on the X chromosome introgression into the ZW populations through emigration of the XY populations. Another model to explain the W chromosome heterogeneity might be proposed. This one is based on a recent bottle neck effect [40]: originally, the W chromosomes were heterogeneous in the ZW populations and then the $W^1$ chromosome has recently reached an extreme homogeneity through geographic isolations and a bottle neck effect, while the $W^{3ab}$ chromosomes still conserve the heterogeneity. However, this model could not explain the discordance between mitochondrial and nuclear genomes in the ZW3b sub-group. Moreover, if it is a recent bottle neck, the theory cannot explain the fact that the degenerated lethal genes to arrest development of WW embryos are not shared between the $W^1$ and $W^{3a}$ chromosomes. In addition, a prerequisite is that no W chromosome degeneration proceeded before separation of the three ZW sub-groups but occurred rapidly and individually after the separations. Or another model might be of ghost admixture to explain the genetic heterogeneity of ZW3 sub-groups [40], but we have never known any ZW population (a ghost) of which genomic constitution (on chromosomes or allozymes) is unique or exceptional in the ZW group. Therefore, our theory based on the sex chromosome recycling is the best candidate to explain the W chromosome heterogeneity in *G. rugosa* that we can propose at present.

### 4.4. Novel Evolutionary Theory on the Origin of Neo-ZW Group

Based on its phylogenetic origin, we had hypothesised in a previous study that the Neo-ZW group had originated by hybridisation between the XY group (with heteromorphic sex chromosomes) and the West-Japan group (with homomorphic XX-XY type of sex chromosomes) [21]. In fact, the mitochondrial haplogroups of the Neo-ZW group are certainly different from those of the ZW group and instead, share homology with those of the XY group. In addition, based on allozymes, the nuclear genomes of the Neo-ZW group supported hybridisation, including that with the West-Japan group [41]. However, the following two findings on hybridisation between different sex determining and sex chromosome systems in this frog species require a revision of the previous hypothesis: (1) in sex determination, female heterogamety dominates male heterogamety [20], and (2) in sex chromosomes, homomorphy dominates heteromorphy [19]. In addition, the present study revealed that the Neo-ZW group evolved by emigration from the ZW3 sub-group into the XY group that was originally located in the present region of the Neo-ZW group. It has been proven experimentally that reproductive isolation is not established amongst the geographic groups of this species [42–44], and that geographic barriers are unconfirmed around the boundaries between the ZW3b sub-group and Neo-ZW group. In addition, hybridisation between the two different species of medaka fish has been reported in the regions corresponding to the boundary between the ZW3b sub-group and Neo-ZW group (*Oryzias sakaizumii* from north and *Oryzias latipes* from the south) [45]. Thus, we reconsidered the involvement of the ZW group in the evolution of the Neo-ZW group and propose that the evolutionary origin of the Neo-ZW group was driven mainly

by hybridisation between the ZW (ZW3 sub-group) and XY groups, with slight genetic introgression from the West-Japan group (Figures 5 and 6).

## 5. Conclusions

The geographic ZW group bearing ZZ-ZW heteromorphic sex chromosomes are genetically classified into three sub-groups, ZW1, ZW2, and ZW3 (a and b) based on mitochondrial *cytochrome b* sequences. The high-throughput analyses of nuclear genomes (autosomes as well as sex chromosomes) uncovered the mechanisms responsible for genetic heterogeneity of the W chromosomes in geographic populations. X chromosomes were recycled into new W chromosomes via recombination between the W and X chromosomes in WX female hybrids (hybrids between ZW females of the ZW3 sub-group and XY males of the XY group). Putting together the results derived from our previous studies in the cases of Neo-ZW1 and Neo-ZW2, it was concluded that the W chromosomes evolved from the X chromosomes independently at least four times in the ZW and Neo-ZW groups. This is a case of intermittent evolution of W chromosomes via interpopulation hybridisation: a repetition of degeneration and resurrection. Finally, one question remains to be answered: Why is female heterogamety always dominant over male heterogamety in this species? The X chromosomes were recycled into W chromosomes in hybrid populations between the sex-chromosome differentiated groups, even though the original W chromosomes were removed from the hybrid populations due to WW mortality and the WY hybrid could be differentiated into a male or a female [20,43]. This seems to contradict the previously reported theory that there is no significant bias in the transition rate between XY and ZW systems in amphibians [46,47]. The evolutionary significance of heterogametic sex choice or epistasis may still be an issue to be resolved to further the understanding of the evolution of sex-determining mechanisms and sex chromosomes.

**Supplementary Materials:** The following supporting information can be downloaded at: https://www.mdpi.com/article/10.3390/dna2030012/s1, Table S1: Autosomal genetic markers (1626 SNPs) isolated from 12 populations, Table S2: Sequences of the autosomal genetic markers (1626 SNPs), Table S3: Sex-linked markers (220 SNPs) isolated from 12 populations, Table S4: Sequences of the sex-linked markers (220 SNPs), Figure S1: Histograms of STRUCTURE assignment test (K = 5, 6 and 7) for autosomal and sex-linked SNPs, Figure S2: Histograms of STRUCTURE assignment test (K = 8, 9 and 10) for autosomal and sex-linked SNPs.

**Author Contributions:** Conceptualization, M.O. and I.M.; methodology, M.O. and T.E.; software, F.S. and M.O.; formal analysis, M.O. and F.S.; resources, M.O. and Y.Y.; writing—original draft preparation, M.O.; writing—review and editing, I.M. and T.E.; funding acquisition, I.M. All authors have read and agreed to the published version of the manuscript.

**Funding:** This research was funded by a grant-in-aid for scientific research from the Ministry of Education, Culture, Sports, Science and Technology of Japan awarded to I.M. (No. 19K06788).

**Institutional Review Board Statement:** Animal care and experimental procedures were approved by the Committee for Ethics in Animal Experimentation at Hiroshima University (Permit Number: G18-2-2).

**Informed Consent Statement:** Not applicable.

**Data Availability Statement:** Mitochondrial *cytochrome b* sequences were deposited in the DDBJ Data Libraries (accession number LC671792-671810).

**Acknowledgments:** We express our sincere thanks to Shintaro Seki for taking and providing the photo of *Glandirana rugosa* used in the graphic abstract.

**Conflicts of Interest:** The authors declare no conflict of interest. The funders had no role in the design of the study; in the collection, analyses, or interpretation of data; in the writing of the manuscript, or in the decision to publish the results.

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
