# Peer review of "W Chromosome Evolution by Repeated Recycling in the Frog Glandirana rugosa"

_2673-8856, doi:10.3390/dna2030012_

Round 1

Reviewer 1 Report

Major comment: I appreciate that the authors took my suggestions to heart and reinstated their earlier data that shows ambiguity to their conclusions. While this makes the story less straight forward, it shows the true nature of what this data can and cannot tell us. I don’t think that the authors’ interpretation of the data is necessarily wrong, but the uncertainty that exists at this point should be acknowledged. I also note that the phylogenetic analysis using nuclear SNPs lends some credit to the authors' hypothesis, as ZW3b clusters with Neo-ZW2, although it doesn’t explain the blue (K=3, ZZ part) in ZW2, given that ZW1 and ZW2 consistently cluster together. Again, there is some ambiguity in the data, but I appreciate that it is all put forth for the reader to draw their own conclusions. Also, the authors’ hypothesis is an exciting one that I hope will provoke some debate.

Line 286: typo in diffirentiation, should be differentiation

Line 288: typo in mechanims, should be mechanisms

Line 288: typo in heterogenity, should be heterogeneity

Line 390: typo in heterogenious, should be heterogeneous

Author Response

Dear reviewer 1,

Thank you for the comments. We revised the spelling errors you mentioned.

Major comment: I appreciate that the authors took my suggestions to heart and reinstated their earlier data that shows ambiguity to their conclusions. While this makes the story less straight forward, it shows the true nature of what this data can and cannot tell us. I don’t think that the authors’ interpretation of the data is necessarily wrong, but the uncertainty that exists at this point should be acknowledged. I also note that the phylogenetic analysis using nuclear SNPs lends some credit to the authors' hypothesis, as ZW3b clusters with Neo-ZW2, although it doesn’t explain the blue (K=3, ZZ part) in ZW2, given that ZW1 and ZW2 consistently cluster together. Again, there is some ambiguity in the data, but I appreciate that it is all put forth for the reader to draw their own conclusions. Also, the authors’ hypothesis is an exciting one that I hope will provoke some debate.

Line 286: typo in diffirentiation, should be differentiation

Line 288: typo in mechanims, should be mechanisms

Line 288: typo in heterogenity, should be heterogeneity

Line 390: typo in heterogenious, should be heterogeneous

Best

Ikuo Miura

Reviewer 2 Report

This is a highly interesting manuscript presenting an unusual trend of sex chromosome evolution in a Japanese frog, Glandirana rugosa which may provide important insight to the evolution of sex chromosomes among lower vertebrates. I have followed the work from this reputed lab with great interest since the original publication of dynamic sex chromosomes transition in Rana rugosa. In this manuscript the authors report intermittent evolution of W chromosomes caused by inter-population hybridisation in the same species now designated as Glandirana rugosa.

First the authors were able to identify different W chromosome clades with mitochondrial gene analysis and they nicely demonstrate that the W chromosomes evolved from the X chromosomes independently at least four times in the ZW and Neo-ZW groups. I think this is a very elegant observation and certainly provides an interesting twist to the conventional mechanism underlying the sex chromosome evolution. I think the work has been meticulously carried out with appropriate tools and the results are properly interpreted.

The authors may consider following suggestions:

-          It might be necessary to present homomorphic and heteromorphic sex chromosomes so that the general readers can follow this interesting piece of work.

-          Is the X chromosome is structurally more similar to the W compared to the Y ?

-          The work of Hillis and Green (1990) may be referred, who identified seven heterogametic transitions (i.e., transitions from male to female heterogamety or vice versa) during amphibian evolutionary history.

Author Response

Dear reviewer 2,

Thank you for the comments. We responded to each of your comments as follows.

Best

Ikuo Miura

This is a highly interesting manuscript presenting an unusual trend of sex chromosome evolution in a Japanese frog, Glandirana rugosa which may provide important insight to the evolution of sex chromosomes among lower vertebrates. I have followed the work from this reputed lab with great interest since the original publication of dynamic sex chromosomes transition in Rana rugosa. In this manuscript the authors report intermittent evolution of W chromosomes caused by inter-population hybridisation in the same species now designated as Glandirana rugosa.

First the authors were able to identify different W chromosome clades with mitochondrial gene analysis and they nicely demonstrate that the W chromosomes evolved from the X chromosomes independently at least four times in the ZW and Neo-ZW groups. I think this is a very elegant observation and certainly provides an interesting twist to the conventional mechanism underlying the sex chromosome evolution. I think the work has been meticulously carried out with appropriate tools and the results are properly interpreted.

The authors may consider following suggestions:

-          It might be necessary to present homomorphic and heteromorphic sex chromosomes so that the general readers can follow this interesting piece of work. 

We added short explanation about the morphology of the X, Y, W and Z chromosomes in introduction section as follows in line 59-62:

“The sex chromosomes in all three groups are represented by homologous chromosome 7 in the 13 haploid complements: the X and W chromosomes are metacentric, sharing one origin, while the Y and Z chromosomes are subtelocentric, sharing the other origin [16-18].”

-          Is the X chromosome is structurally more similar to the W compared to the Y ?

Yes it is. The explanation above added in the introduction tells you the answer.

-          The work of Hillis and Green (1990) may be referred, who identified seven heterogametic transitions (i.e., transitions from male to female heterogamety or vice versa) during amphibian evolutionary history.

Thanks. We referred to two recent papers of Pennel et al. and Evans et al. in line 450, because much more data on heterogametic sex in amphibians were accumulated since the time of Hillis and Green and the interpretation about direction of heterogametic sex transition is a bit changed.

This manuscript is a resubmission of an earlier submission. The following is a list of the peer review reports and author responses from that submission.

Round 1

Reviewer 1 Report

The manuscript of Ogata et al. is devoted to the study of peculiar sex chromosome evolution in Glandirana rugosa, a frog species famous by having different sex chromosome systems in different geographic populations. In the current work the authors show how intra-specific hybridization "revived" the degenerate W chromosomes of the ZW lineages by recombination with the X chromosome of the XY lineages.

The only disadvantage of this manuscript is that it requires detailed knowledge of the "previous episodes" for understanding. For example, it is not clear for me for what purpose the authors needed to check the microsatellite upstream to Sox3. The results of this test are not discussed in detail. I suggest significantly expanding the Introduction section to tell more about the results of the previous studies, about the methods accepted in the G. rugosa studies (why are the fluorophores required during the amplification of the microsatellite, etc).

Minor issues:

Line 271, capital I needed.

Line 297, "W-borne Ar"

Author Response

Dear reviewer 1,

We have responded all the comments by the reviewer 1 point to point and in according to them, we revised the text and figures. The revised words and sentences in the text are written in blue. Our responses to the comments are written in blue as below. Thank you so much for pointing the important parts.

Reviewer 1

The manuscript of Ogata et al. is devoted to the study of peculiar sex chromosome evolution in Glandirana rugosa, a frog species famous by having different sex chromosome systems in different geographic populations. In the current work the authors show how intra-specific hybridization "revived" the degenerate W chromosomes of the ZW lineages by recombination with the X chromosome of the XY lineages.

The only disadvantage of this manuscript is that it requires detailed knowledge of the "previous episodes" for understanding. For example, it is not clear for me for what purpose the authors needed to check the microsatellite upstream to Sox3. The results of this test are not discussed in detail. I suggest significantly expanding the Introduction section to tell more about the results of the previous studies, about the methods accepted in the G. rugosa studies (why are the fluorophores required during the amplification of the microsatellite, etc).

Thanks for the comments that Introduction about the story of sex chromosomes in G. rugosa and results and discussion about sox3 upstream microsattelite are not enough.

・We added evolutionary story of Neo-ZW to the introduction section ( L58-68 ) as below:

“The Neo-ZW group has a recent origin from XY group and is divided into two sub-groups of Neo-ZW1 and Neo-ZW2 based on the mitochondrial haplotypes: the latter is proved to be derived from more recent hybridization between Neo-ZW1 sub-group and XY group (Ogata et al., 2018).”

・The sox3 microsatellite haplotypes indicate that the haplotypes originated in XY populations such as 248 and 240 are deeply introgressed into ZW3 sub-group, supporting the introdluction  theory of XY populations into ZW populations based on structure maps (blue allele numbers in Table 1). Therefore, we added sentences in results and discussion (L251-252 and L301) as below.

L251-252:

“These alleles support X chromosome introduction from XY into ZW3 populations.”

L299:

“Second, we found that the sex chromosomes of the ZW (three sub-groups) and Neo-ZW groups (two sub-groups) were genomically classified into six types of Z chromosomes (Z1, Z2, Z3a, Z3b, ZN1, and ZN2) and five types of W chromosomes (W1, W3a, W3b, WN1, and WN2), respectively, based on sex-linked SNPs and Sox3 upstream microsatellite, W-Ar expression, and WW embryo mortality (Figure 4).”

・Fluorochrome is attached to the primer sequence in order to identify the amplified fragments electrophoresed on the sequencing machine (ABI Prism 310 genetic analyzer).

Minor issues:

Line 271, capital I needed.

Revised to “Ohi-Itoigawa” in L284 and picture of figure 5.

Line 297, "W-borne Ar"

Thanks. Revised.

Reviewer 2 Report

Main Concern: The STRUCTURE analysis gives us some idea but may not be sufficient to deduce that certain introgression events have taken place. The deductions based on the STRUCTURE plots at K=3 would not be made if one were to look at K=4. Two examples: 1) The autosomal “introgression event” from XY to ZW3a changes from blue (K=3) to green (K=4), which indicates that there this portion of genetic variation is closer to either Neo-ZW2 or ZW3b than to XY. 2) The XY/ZW “introgression event” from XY to ZW2 changes from blue (K=3) to red (K=4) and hence completely disappears. These two examples highlight the problems of deducing population admixture models from STRUCTURE plots. I recommend additional analyses that would more strongly support deduced events. A phylogenetic analysis of the genetic sequences that are hypothesized to be involved in introgression events could resolve the relationships between those sequences.

Other concerns:

I like the idea behind Fig. 4, but it would be good if it was explained more explicitly. The introgression hypotheses are quite detailed not easily understood just by studying the figure.

L. 122, I was not able to find a dataset under the accession LC671792-671810. Will this dataset be available upon publication?

L. 206, Is it correct that it says K=5 and K=6 instead of K=4? Given that Fig. 3 shows K=4, should not that value be reported?

Fig. 4, Why is Neo-ZW2 missing from the upper panel?

L. 298-300. “the Z2 chromosomes of ZW2 are almost entirely replaced with Y chromosomal segments of the XY group.”; L. 306, “W3b mostly comprises parts of X chromosomes”; L. 308-311, “Z3a chromosomes of ZW3a are almost entirely replaced with Y chromosomes (Figures 4, 5B-❶), while major parts of Z3b have accepted introgression from ZN1 of Neo-ZW1 sub-groups (Figures 4, 5C-❸a).”

The data underlying these claims is not shown as far as I can see. Figure 4 does not discriminate in color between X and Y with regard to introgression from XY to Z2, for instance, similar for all the other cases cited here. Figure 5 is a schematic that summarizes a hypothesis (as is Figure 4, lower panel), but does not present original data. This data needs to be presented somehow. It needs to be shown that Z2 sequences stem from Y (possibly via Z3a), that W3b sequence comes from the X of XY, and so on. It could for instance be done using phylogenetic analysis of sequences from the chromosomes involved.

Minor things, typos etc.

- Fig. 1, In the Figure legend, “Glandirana is misspelled.

- Fig. 2, The annotation “ZW1” (I believe) in K=3, Autosomes, is not readable, please add a white background or make it otherwise visible.

- Fig. 4, The annotation R1-R4 is not correctly displayed in the PDF, but rendered as white boxes.

- L. 289, I suggest changing “repetitious” to “repeated”

- L. 297, “Ar-borne W” should be “W-borne Ar”

- L. 383, “epistatic” should be “epistasis”

- L.387, “Glandirana is misspelled again in the Acknowledgments.

Author Response

Dear reviewer 2,

We have responded all the comments by the reviewer point to point and in according to them, we revised the text and figures. The revised words and sentences in the text are written in blue. The figures 3, 4 and 5 are replaced with revised ones. Our responses to the comments are written in blue as below. Thank you so much for pointing the important parts.

Reviewer 2

Main Concern: The STRUCTURE analysis gives us some idea but may not be sufficient to deduce that certain introgression events have taken place. The deductions based on the STRUCTURE plots at K=3 would not be made if one were to look at K=4. Two examples: 1) The autosomal “introgression event” from XY to ZW3a changes from blue (K=3) to green (K=4), which indicates that there this portion of genetic variation is closer to either Neo-ZW2 or ZW3b than to XY. 2) The XY/ZW “introgression event” from XY to ZW2 changes from blue (K=3) to red (K=4) and hence completely disappears. These two examples highlight the problems of deducing population admixture models from STRUCTURE plots. I recommend additional analyses that would more strongly support deduced events. A phylogenetic analysis of the genetic sequences that are hypothesized to be involved in introgression events could resolve the relationships between those sequences.

Great thanks to the comments detecting good points.

As the reviewer said, STRUCTURE might not the best tool for seeing population dynamics here, but we chose K=3 maps for the discussion because of the following three reasons:

  • Delta K values are higher in K=2 and 3 than 4 and others. K=2 value is the largest, but not largely far from K=3. On the other hand, K=4 and others are very lower and is hard to be supported.
  • K=3 clearly discriminates the genomes of three (sub-) groups, XY, Neo-ZW1 and ZW1 in blue, yellow and red, respectively.
  • The autosomal hybrid origin of Noe-ZW2 was clearly indicated in K=3 in blue and yellow. (We added the story of the origin of Neo-ZW2 in introduction section in L58-68 and the text in L215-219).

Even if we use phylogenetic analysis (tree), we would meet similar or complicated situation because of the low bootstrap values at the nodes. Therefore, the K=3 map is the best tool we can choose here in our study. Of course, we are planning to deeply investigate the population dynamics using whole genome sequencing to confirm our results and theroy, but it takes a bit more time.

As reviewer pointed, the colors of some parts in K=3 are changed from blue to green or blue to red in K=4. These are not significantly supported because of their lower delta K values: delta K (K=4) are 5.82 (autosomal), 6.95 (ZZ) and 1. 38 (heterozygous, XY or ZW). Particularly, green looks like indicating genomic feature of Neo-ZW2 in K=4, but this sub-group has a more recent origin at hybridization between Neo-ZW1 and XY and composed of the two distinct genomes. The green color cluster in K=4 makes the discrimination between XY and ZW genomes less clear. Therefore, we deleted the K4 maps from figure 3.

Other concerns:

I like the idea behind Fig. 4, but it would be good if it was explained more explicitly. The introgression hypotheses are quite detailed not easily understood just by studying the figure.

  1. 122, I was not able to find a dataset under the accession LC671792-671810. Will this dataset be available upon publication?

Yes, it will be open automatically in six months or soon just after publication.

  1. 206, Is it correct that it says K=5 and K=6 instead of K=4? Given that Fig. 3 shows K=4, should not that value be reported?

These are our mistakes. The delta values are right but their maps are not put in figure 3.  We have changed the sentence as below and deleted the K4 maps.

L210-211: “Delta K values for heterogametic frogs were 24.60 (k=2), 17.21 (k=3), and 1.38 (k=4), while those for homogametic ZZ frogs were 35.21 (k=2), 21.94 (k=3), and 6.95 (k=4).”

Fig. 4, Why is Neo-ZW2 missing from the upper panel?

Neo-ZW2 was added on the upper panel in figure 4.

  1. 298-300. “the Z2chromosomes of ZW2 are almost entirely replaced with Y chromosomal segments of the XY group.”; L. 306, “W3bmostly comprises parts of X chromosomes”; L. 308-311, “Z3a chromosomes of ZW3a are almost entirely replaced with Y chromosomes (Figures 4, 5B-❶), while major parts of Z3b have accepted introgression from ZN1 of Neo-ZW1 sub-groups (Figures 4, 5C-❸a).”

The data underlying these claims is not shown as far as I can see. Figure 4 does not discriminate in color between X and Y with regard to introgression from XY to Z2, for instance, similar for all the other cases cited here. Figure 5 is a schematic that summarizes a hypothesis (as is Figure 4, lower panel), but does not present original data. This data needs to be presented somehow. It needs to be shown that Z2 sequences stem from Y (possibly via Z3a), that W3b sequence comes from the X of XY, and so on. It could for instance be done using phylogenetic analysis of sequences from the chromosomes involved.

We tried to discriminate X from Y in Structure maps by comparing both maps of heterozygotes (ZW or XY) and homozygotes (ZZ), and particularly Neo-ZW2 is available. As Neo-ZW2 is a new sub-group that has a more recent origin at hybridization between Neo-ZW1 and XY, we can see the hybrid genome composition in Neo-ZW2: the W chromosomes originated mainly from X (yellow) while Z chromosomes equally from ZNeo1(yellow) and Y (blue) in the structure maps (also, see Ogata et al., 2018): here we can discriminate Y from ZNeo1 (this Y segment is indicated by “Y” in ZZ males map (K=3) of figure 4). Thus, based on the maps, we concluded that Z chromosomes of ZW2 and ZW3a are mostly replaced with Y chromosomes (blue), and Z chromosomes of ZW3b are mostly with ZNeo1(yellow), which we can see in the K=3 structure maps (ZW/XY and ZZ).

The sentences including revisions are shown below (L207-222 and L298-324).

L207-222:

“To investigate the genomic structure of the sex chromosomes in the ZW populations, separate allele distributions were created for homogametic ZZ males as well as for heterogametic ZW females and XY males, in structure maps of 220 sex-linked SNPs (Tables S3 and S4). Delta K values for heterogametic frogs were 24.60 (k=2), 17.21 (k=3), and 1.38 (k=4), while those for homogametic ZZ frogs were 35.21 (k=2), 21.94 (k=3), and 6.95 (k=4). In the K3 map of heterogametic frogs, XY alleles constituted half or less of ZW females of the ZW3a or b sub-groups. In the homogametic ZZ frog map, Y alleles (blue indicates Y alleles while yellow indicates Z alleles of Neo-ZW1 in ZZ males of Neo-ZW2, because this sub-group has a more recent origin at hybridization between XY and Neo-ZW1, see Ogata et al., 2018) constituted one-third of ZZ males from ZW3b, and almost all of ZZ males from ZW3a and ZW2 (arrows in K=3 map of ZZ chromosomes in Figure 3 and Figure 4). “

L298-324:

“Second, we found that the sex chromosomes of the ZW (three sub-groups) and Neo-ZW groups (two sub-groups) were genomically classified into six types of Z chromosomes (Z1, Z2, Z3a, Z3b, ZN1, and ZN2) and five types of W chromosomes (W1, W3a, W3b, WN1, and WN2), respectively, based on sex-linked SNPs and Sox3 upstream microsatellite, W-Ar expression, and WW embryo mortality (Figure 4). The evolutionary mechanisms of the Z and W chromosomes in Neo-ZW1 and Neo-ZW2 sub-groups have been described in our previous studies [20, 21]. In this study, the W1 chromosomes in the ZW group were found to be shared by the ZW1 and ZW2 sub-groups: W-borne Arwas not expressed and WW embryos died at 10 dpf with oedema (L10 in Figure 4). On the other hand, their Z chromosomes are not shared: the Z2 chromosomes of ZW2 are almost entirely replaced with Y chromosomal segments of the XY group, as ZZ clusters of ZW2 sub-group are mostly indicated in blue, which are the Y chromosomal segments of Neo-ZW2 in K3 map of Figure 3). Although autosomal SNPs did not indicate genomic introgression of the XY group into the ZW2, the genomic region of the Y chromosomes might have been introduced via XY males following separation from ZW1, and then may have been replaced with the Z1 chromosome (Figures 4, 5B-❷). The genetic structures of the ZW sex chromosomes differ from each other between the ZW3a and 3b sub-groups: the major part of W3a chromosome mostly comprises the original W1 chromosome except for the Ar or lethal genes, whereas W3b mostly comprises parts of X chromosomes, as ZZ clusters of ZW3b are mostly indicated in yellow (ZNeo1) while ZW clusters are equally both in yellow and blue showing that the W3b chromosomes originated from X chromosomes (blue) (in K3 map of Figure 3). W3ab-borne Ar may or may not be expressed, and W3aW3a embryos die at 40 dpf owing to underdevelopment (L40). Experimental data for the development of W3bW3b embryos is not available. Z3a chromosomes of ZW3a are almost entirely replaced with Y chromosomes (shown in blue in K3 map of Figure 3, and Figure 5B-❶), while major parts of Z3b have accepted introgression from ZN1 of Neo-ZW1 sub-groups (shown in yellow in K3 map of Figure 4 and Figure 5C-❸a). “

Minor things, typos etc.

- Fig. 1, In the Figure legend, “Glandirana” is misspelled.

Thanks. Revised.

- Fig. 2, The annotation “ZW1” (I believe) in K=3, Autosomes, is not readable, please add a white background or make it otherwise visible.

“XY” and “ZW” are described in K3 maps in figure 3. They mean XY group and ZW group, respectively.

- Fig. 4, The annotation R1-R4 is not correctly displayed in the PDF, but rendered as white boxes.

Thanks.  We will check them at a final version.

- L. 289, I suggest changing “repetitious” to “repeated”

Thanks. Revised to “repeated” in the text and title.

- L. 297, “Ar-borne W” should be “W-borne Ar”

Our terrible mistake. Revised.

- L. 383, “epistatic” should be “epistasis”

Revised.

- L.387, “Glandirana” is misspelled again in the Acknowledgments.

Revised.

Round 2

Reviewer 2 Report

Instead of addressing my main concern with the study, the authors have obscured data that contradicts their favored hypothesis and disregarded my issues with it. I can imagine that this favored hypothesis is indeed true. However, apart from this procedure being ethically dubious, it is concerning for a number of reasons specific to the situation at hand.

First, STRUCTURE and related methods are hypothesis-generating methods. This means that their results are the starting points for a study, not the final results. The hypotheses that were constructed using the STRUCTURE analysis need to be confirmed using actual phylogenetic analyses of appropriate genetic markers on the chromosomes involved. This is complicated and time consuming, but this is the only way of knowing if the inferred STRUCTURE models are actually reflecting reality.

Second, a chosen K is never the “correct K” – there is no correct K, which means that alternative possibilities need to be considered, while the authors instead chose to ignore them. I refer the authors to Lawson et al. Nat. Commun. 2018 for a discussion on the pitfalls of choosing K. Importantly, the value of K chosen by any method is often an underestimate, as it is the smallest K that is able to explain most of the data. This does not mean that all of the data is explained by this K. Especially in cases where the authors highlight small parts of variation that they claim may have introgressed from A to B, this small part of the variation may not conform to the model proposed by K=3. This means that alternative possibilities need to be considered and mentioned in the discussion or these alternatives need to be ruled out by additional analyses that are able to do that. STRUCTURE is no such analysis.

Third, choosing a certain value for K is generally interpreted as it representing the ancestral number of populations prior to a recent admixture event. However, this simple model is not able to deal with isolation by distance-type scenarios, nested population structures, or changes in population structure throughout longer time periods. Crucially, all of these things appear to have occurred in G. rugosa, indicating that the inferred K and associated model are less reliable than under a simpler demographic history. In other words, several of the basic assumptions of this STRUCTURE analysis do not hold, and hence, the results of the analysis should be treated with great caution.

I think the authors need to do either of two things (preferably both): 1) Confirm their STRUCTURE-generated hypotheses using appropriate phylogenetic analyses and 2) discuss all the caveats and pitfalls, including unmet model assumptions with great care.